# Graph Information Bottleneck

**Tailin Wu**[*], **Hongyu Ren**[*]**, Pan Li, Jure Leskovec**
Department of Computer Science
Stanford University
{tailin, hyren, panli0, jure}@cs.stanford.edu

## Abstract

Representation learning of graph-structured data is challenging because both graph structure and node features carry important information. Graph Neural Networks (GNNs) provide an expressive way to fuse information from network structure and node features. However, GNNs are prone to adversarial attacks. Here we introduce *Graph Information Bottleneck (GIB)*, an information-theoretic principle that optimally balances expressiveness and robustness of the learned representation of graph-structured data. Inheriting from the general Information Bottleneck (IB), GIB aims to learn the minimal sufficient representation for a given task by maximizing the mutual information between the representation and the target, and simultaneously constraining the mutual information between the representation and the input data. Different from the general IB, GIB regularizes the *structural* as well as the *feature* information. We design two sampling algorithms for structural regularization and instantiate the GIB principle with two new models: GIB-Cat and GIB-Bern, and demonstrate the benefits by evaluating the resilience to adversarial attacks. We show that our proposed models are more robust than state-of-the-art graph defense models. GIB-based models empirically achieve up to 31% improvement with adversarial perturbation of the graph structure as well as node features.

## 1 Introduction

Representation learning on graphs aims to learn representations of graph-structured data for downstream tasks such as node classification and link prediction [1, 2]. Graph representation learning is a challenging task since both node features as well as graph structure carry important information [3, 4]. Graph Neural Networks (GNNs) [1, 3, 5–7] have demonstrated impressive performance, by learning to fuse information from both the node features and the graph structure [8].

Recently, many works have been focusing on developing more powerful GNNs [8–13], in a sense that they can fit more complex graph-structured data. However, at present GNNs still suffer from a few problems. For example, the features of a neighborhood node can contain non-useful information that may negatively impact the prediction of the current node [14]. Also, GNN's reliance on message passing over the edges of the graph also makes it prone to noise and adversarial attacks that target at the graph structure [15, 16].

Here we address the above problems and rethink what is a "good" representation for graph-structured data. In particular, the Information Bottleneck (IB) [18, 19] provides a critical principle for representation learning: an optimal representation should contain the *minimal sufficient* information for the downstream prediction task. IB encourages the representation to be maximally informative about the target to make the prediction accurate (*sufficient*). On the other hand, IB also discourages the representation from acquiring additional information from the data that is irrelevant for predicting the

---

[*]Equal contribution

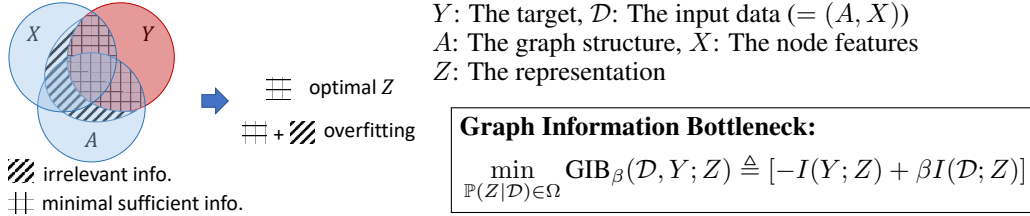

$Y$: The target, $\mathcal{D}$: The input data $(=(A, X))$
$A$: The graph structure, $X$: The node features
$Z$: The representation

**Graph Information Bottleneck:**

$$\min_{\mathbb{P}(Z|\mathcal{D})\in\Omega} \text{GIB}_\beta(\mathcal{D}, Y; Z) \triangleq [-I(Y; Z) + \beta I(\mathcal{D}; Z)]$$

Figure 1: Graph Information Bottleneck is to optimize the representation $Z$ to capture the minimal sufficient information within the input data $\mathcal{D} = (A, X)$ to predict the target $Y$. $\mathcal{D}$ includes information from both the graph structure $A$ and node features $X$. When $Z$ contains irrelevant information from either of these two sides, it overfits the data and is prone to adversarial attacks and model hyperparameter change. $\Omega$ defines the search space of the optimal model $\mathbb{P}(Z|\mathcal{D})$. $I(\cdot; \cdot)$ denotes the mutual information [17].

target (*minimal*). Based on this learning paradigm, the learned model naturally avoids overfitting and becomes more robust to adversarial attacks.

However, extending the IB principle to representation learning on graph-structured data presents two unique challenges. First, previous models that leverage IB assume that the training examples in the dataset are independent and identically distributed (i.i.d.). For graph-structured data, this assumption no longer holds and makes model training in the IB principle hard. Moreover, the structural information is indispensable to represent graph-structured data, but such information is discrete and thus hard to optimize over. How to properly model and extract minimal sufficient information from the graph structure introduces another challenge that has not been yet investigated when designing IB-based models.

We introduce Graph Information Bottleneck (GIB), an information-theoretic principle inherited from IB, adapted for representation learning on graph-structured data. GIB extracts information from both the graph structure and node features and further encourages the information in learned representation to be both minimal and sufficient (Fig. 1). To overcome the challenge induced by non-i.i.d. data, we further leverage local-dependence assumption of graph-structure data to define a more tractable search space $\Omega$ of the optimal $\mathbb{P}(Z|\mathcal{D})$ that follows a Markov chain to hierarchically extract information from both features and structure. To our knowledge, our work provides the first information-theoretic principle for supervised representation learning on graph-structured data.

We also derive variational bounds for GIB, making GIB tractable and amenable for the design and optimization of GNNs. Specifically, we propose a variational upper bound for constraining the information from the node features and graph structure, and a variational lower bound for maximizing the information in the representation to predict the target.

We demonstrate the GIB principle by applying it to the Graph Attention Networks (GAT) [5], where we leverage the attention weights of GAT to sample the graph structure in order to alleviate the difficulty of optimizing and modeling the discrete graph structure. We also design two sampling algorithms based on the categorical distribution and Bernoulli distribution, and propose two models GIB-Cat and GIB-Bern. We show that both models consistently improve robustness w.r.t. standard baseline models, and outperform other state-of-the-art defense models. GIB-Cat and GIB-Bern improve the classification accuracy by up to 31.3% and 34.0% under adversarial perturbation, respectively.
Project website and code can be found at `http://snap.stanford.edu/gib/`.

## 2 Preliminaries and Notation

**Graph Representation Learning**. Consider an undirected attributed graph $G = (V, E, X)$ with $n$ nodes, where $V = [n] = \{1, 2, ...n\}$ is the node set, $E \subseteq V \times V$ is the edge set and $X \in \mathbb{R}^{n \times f}$ includes the node attributes. Let $A \in \mathbb{R}^{n \times n}$ denote the adjacency matrix of $G$, *i.e.*, $A_{uv} = 1$ if $(u, v) \in E$ or 0 otherwise. Also, let $d(u, v)$ denote the shortest path distance between two nodes $u, v (\in V)$ over $A$. Hence our input data can be overall represented as $\mathcal{D} = (A, X)$.

In this work, we focus on node-level tasks where nodes are associated with some labels $Y \in [K]^n$. Our task is to extract node-level representations $Z_X \in \mathbb{R}^{n \times f'}$ from $\mathcal{D}$ such that $Z_X$ can be further

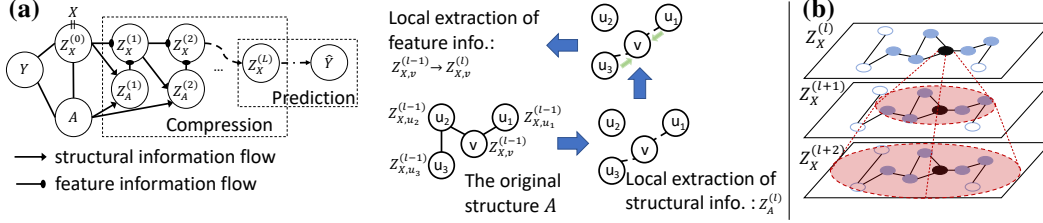

Figure 2: Our GIB principle leverages local-dependence assumption. **(a)** The Markov chain defines the search space $\Omega$ of our GIB principle, of which each step uses a local-dependence assumption to extract information from the structure and node features. The correlation between node representations are established in a hierarchical way: Suppose local dependence appears within 2-hops given the structure $A$. **(b)** In the graph, given the representations $Z_X^{(l)}$ of the blue nodes and $A$ that conveys the structural information that the blue nodes lie within 2-hops of the black node, the representations $Z_X^{(l+1)}$ are independent between the black node and the white nodes. However, the correlation between them may be established in $Z_X^{(l+2)}$.

used to predict $Y$. We also use the subscript with a certain node $v \in V$ to denote the affiliation with node $v$. For example, the node representation of $v$ is denoted by $Z_{X,v}$ and its label is denoted by $Y_v$.

**Notation**. We do not distinguish the notation of random variables and of their particular realizations if there is no risk of confusion. For any set of random variables $H$, we use $\mathbb{P}(H)$, $\mathbb{Q}(H)$, ... to denote joint probabilistic distribution functions (PDFs) of the random variables in $H$ under different models. $\mathbb{P}(\cdot)$ corresponds to the induced PDF of the proposed model while $\mathbb{Q}(H)$ and $\mathbb{Q}_i(H)$, $i \in \mathbb{N}$ correspond to some other distributions, typically variational distributions. For discrete random variables, we use generalized PDFs that may contain the Dirac delta functions [20]. In this work, if not specified, $\mathbb{E}[H]$ means the expectation over all the random variables in $H$ w.r.t. $\mathbb{P}(H)$. Otherwise, we use $\mathbb{E}_{\mathbb{Q}(H)}[H]$ to specify the expectation w.r.t. other distributions denoted by $\mathbb{Q}(H)$. We also use $X_1 \perp X_2 | X_3$ to denote that $X_1$ and $X_2$ are conditionally independent given $X_3$. Let $\text{Cat}(\phi)$, $\text{Bernoulli}(\phi)$ denote the categorical distribution and Bernoulli distribution respectively with parameter $\phi$ ($\in \mathbb{R}_{\geq 0}^{1 \times C}$). For the categorical distribution, $\phi$ corresponds to the probabilities over different categories and thus $\|\phi\|_1 = 1$. For the Bernoulli distribution, we generalize it to high dimensions and assume we have $C$ independent components and each element of $\phi$ is between 0 and 1. Let $\text{Gaussian}(\mu, \sigma^2)$ denote the Gaussian distribution with mean $\mu$ and variance $\sigma^2$. $\mu$ and $\sigma^2$ could be vectors with the same dimension, in which case the Gaussian distribution is with the mean vector $\mu$ and covariance matrix $\Sigma = \text{diag}(\sigma^2)$. Let $\Phi(\cdot : \mu, \sigma^2)$ denote its PDF. We use $[i_1 : i_2]$ to slice a tensor w.r.t. indices from $i_1$ to $i_2 - 1$ of its last dimension.

## 3 Graph Information Bottleneck

### 3.1 Deriving the Graph Information Bottleneck Principle

In general, the graph information bottleneck (GIB) principle, inheriting from the principle of information bottleneck (IB), requires the node representation $Z_X$ to minimize the information from the graph-structured data $\mathcal{D}$ (compression) and maximize the information to $Y$ (prediction). However, optimization for the most general GIB is challenging because of the correlation between data points. The i.i.d. assumption of data points is typically used to derive variational bounds and make accurate estimation of those bounds to learn IB-based models [21, 22]. However, for the graph-structured data $\mathcal{D}$, this is impossible as node features, *i.e.*, different rows of $X$, may be correlated due to the underlying graph structure $A$. To fully capture such correlation, we are not allowed to split the whole graph-structured data $\mathcal{D}$ w.r.t. each node. In practice, we typically have only a large network, which indicates that only one single realization of $\mathbb{P}(\mathcal{D})$ is available. Hence, approximating the optimal $Z_X$ in the general formulation GIB seems impossible without making additional assumptions.

Here, we rely on a widely accepted *local-dependence* assumption for graph-structured data: Given the data related to the neighbors within a certain number of hops of a node $v$, the data in the rest of the graph will be independent of $v$. We use this assumption to constrain the space $\Omega$ of optimal representations, which leads to a more tractable GIB principle. That is, we assume that the optimal representation follows the Markovian dependence shown in Fig. 2. Specifically, $\mathbb{P}(Z_X|\mathcal{D})$ iterates node representations to hierarchically model the correlation. In each iteration $l$, the local-dependence assumption is used: The representation of each node will be refined by incorporating its neighbors

w.r.t a graph structure $Z_A^{(l)}$. Here, $\{Z_A^{(l)}\}_{1\leq l\leq L}$ is obtained by locally adjusting the original graph structure $A$ and essentially controlling the information flow from $A$. Finally, we will make predictions based on $Z_X^{(L)}$. Based on this formulation, the objective reduces to the following optimization:

$$\min_{\mathbb{P}(Z_X^{(L)}|\mathcal{D})\in\Omega} \text{GIB}_\beta(\mathcal{D},Y;Z_X^{(L)}) \triangleq \left[-I(Y;Z_X^{(L)})+\beta I(\mathcal{D};Z_X^{(L)})\right] \tag{1}$$

where $\Omega$ characterizes the space of the conditional distribution of $Z_X^{(L)}$ given the data $\mathcal{D}$ by following the probabilistic dependence shown in Fig. 2. In this formulation, we just need to optimize two series of distributions $\mathbb{P}(Z_X^{(l)}|Z_X^{(l-1)},Z_A^{(l)})$ and $\mathbb{P}(Z_A^{(l)}|Z_X^{(l-1)},A)$, $l\in[L]$, which have local dependence between nodes and thus are much easier to be parameterized and optimized.

**Variational Bounds.** Even using the reduced GIB principle and some proper parameterization of $\mathbb{P}(Z_X^{(l)}|Z_X^{(l-1)},Z_A^{(l)})$ and $\mathbb{P}(Z_A^{(l)}|Z_X^{(l-1)},A)$, $l\in[L]$, exact computation of $I(Y;Z_X^{(L)})$ and $I(\mathcal{D};Z_X^{(L)})$ is still intractable. Hence, we need to introduce variational bounds on these two terms, which leads to the final objective to optimize. Note that variational methods are frequently used in model optimization under the traditional IB principle [21]. However, we should be careful to derive these bounds as the data points now are correlated. We introduce a lower bound of $I(Y;Z_X^{(L)})$, which is reproduced from [22, 23], and an upper bound of $I(\mathcal{D};Z_X^{(L)})$, as shown in Propositions 3.1 and 3.2.

**Proposition 3.1** (The lower bound of $I(Y;Z_X^{(L)})$). *For any distributions $\mathbb{Q}_1(Y_v|Z_{X,v}^{(L)})$ for $v\in V$ and $\mathbb{Q}_2(Y)$, we have*

$$I(Y;Z_X^{(L)}) \geq 1 + \mathbb{E}\left[\log\frac{\prod_{v\in V}\mathbb{Q}_1(Y_v|Z_{X,v}^{(L)})}{\mathbb{Q}_2(Y)}\right] + \mathbb{E}_{\mathbb{P}(Y)\mathbb{P}(Z_X^{(L)})}\left[\frac{\prod_{v\in V}\mathbb{Q}_1(Y_v|Z_{X,v}^{(L)})}{\mathbb{Q}_2(Y)}\right] \tag{2}$$

**Proposition 3.2** (The upper bound of $I(\mathcal{D};Z_X^{(L)})$). *We choose two groups of indices $S_X,S_A\subset[L]$ such that $\mathcal{D}\perp Z_X^{(L)}|\{Z_X^{(l)}\}_{l\in S_X}\cup\{Z_A^{(l)}\}_{l\in S_A}$ based on the Markovian dependence in Fig. 2, and then for any distributions $\mathbb{Q}(Z_X^{(l)})$, $l\in S_X$, and $\mathbb{Q}(Z_A^{(l)})$, $l\in S_A$,*

$$I(\mathcal{D};Z_X^{(L)}) \leq I(\mathcal{D};\{Z_X^{(l)}\}_{l\in S_X}\cup\{Z_A^{(l)}\}_{l\in S_A}) \leq \sum_{l\in S_A}\text{AIB}^{(l)} + \sum_{l\in S_X}\text{XIB}^{(l)}, \text{where} \tag{3}$$

$$\text{AIB}^{(l)} = \mathbb{E}\left[\log\frac{\mathbb{P}(Z_A^{(l)}|A,Z_X^{(l-1)})}{\mathbb{Q}(Z_A^{(l)})}\right], \text{XIB}^{(l)} = \mathbb{E}\left[\log\frac{\mathbb{P}(Z_X^{(l)}|Z_X^{(l-1)},Z_A^{(l)})}{\mathbb{Q}(Z_X^{(l)})}\right], \tag{4}$$

The proofs are given in Appendix B and C. Proposition 3.2 indicates that we need to select a group of random variables with index sets $S_X$ and $S_A$ to guarantee the conditional independence between $\mathcal{D}$ and $Z_X^{(L)}$. Note that $S_X$ and $S_A$ that satisfy this condition have the following properties: (1) $S_X\neq\emptyset$, and (2) suppose the greatest index in $S_X$ is $l$ and then $S_A$ should contain all integers in $[l+1,L]$.

To use GIB, we need to model $\mathbb{P}(Z_A^{(l)}|Z_X^{(l-1)},A)$ and $\mathbb{P}(Z_X^{(l)}|Z_X^{(l-1)},Z_A^{(l)})$. Then, we choose some variational distributions $\mathbb{Q}(Z_X^{(l)})$ and $\mathbb{Q}(Z_A^{(l)})$ to estimate the corresponding $\text{AIB}^{(l)}$ and $\text{XIB}^{(l)}$ for regularization, and some $\mathbb{Q}_1(Y_v|Z_{X,v}^{(L)})$ and $\mathbb{Q}_2(Y)$ to specify the lower bound in Eq. (2). Then, plugging Eq. (2) and Eq. (3) into the GIB principle (Eq. (1)), one obtains an upper bound on the objective to optimize. Note that any model that parameterizes $\mathbb{P}(Z_A^{(l)}|Z_X^{(l-1)},A)$ and $\mathbb{P}(Z_X^{(l)}|Z_X^{(l-1)},Z_A^{(l)})$ can use GIB as the objective in training. In the next subsection, we will introduce two instantiations of GIB, which is inspired by GAT [5].

## 3.2 Instantiating the GIB Principle

The GIB principle can be applied to many GNN models. As an example, we apply it to the Graph Attention Network model [5] and present GIB-Cat and GIB-Bern. Algorithm 1 illustrates the base framework of both models with different neighbor sampling methods shown in Algorithm 2 and 3. In each layer, GIB-Cat and GIB-Bern need to first refine the graph structure using the attention weights to obtain $Z_A^{(l)}$ (Step 3) and then refines node representations $Z_X^{(l)}$ by propagating $Z_X^{(l-1)}$ over $Z_A^{(l)}$

(Steps 4-7). Concretely, we design two algorithms for neighbor sampling, which respectively use the categorical distribution and the Bernoulli distribution. For the categorical version, we view the attention weights as the parameters of categorical distributions to sample the refined graph structure to extract structural information. We sample $k$ neighbors with replacement from the pool of nodes $V_{vt}$ for each node $v$, where $V_{vt}$ includes the nodes whose shortest-path-distance to $v$ over $A$ is $t$. We use $\mathcal{T}$ as an upper limitation of $t$ to encode the local-dependence assumption of the GIB principle, which also benefits the scalability of the model. For the Bernoulli version, we model each pair of node $v$ and its neighbors independently with a Bernoulli distribution parameterized by the attention weights. Note that here we did not normalize it with the softmax function as in the categorical version, however, we use the sigmoid function to squash it between 0 and 1. Here we do not need to specify the number of neighbors one node sample ($k$ in the categorical version). Step 4 is sum-pooling of the neighbors, and the output will be used to compute the parameters for a Gaussian distribution where the refined node representations will be sampled. Note that we may also use a mechanism similar to multi-head attention [5]: We split $\tilde{Z}_X^{(l-1)}$ into different channels w.r.t. its last dimension, perform Steps 2-7 independently for each channel and then concatenate the output of different channels to obtain new $Z_X^{(l)}$. Moreover, when training the model, we adopt reparameterization trick for Steps 3 and 7: Step 3 uses Gumbel-softmax [24, 25] while Step 7 uses $\hat{Z}_{X,v}^{(l)} = \mu_v^{(l)} + \sigma_v^{(l)} \odot z$ where $z \sim \text{Gaussian}(0, I)$, $z \in \mathbb{R}^{1 \times f'}$ and $\odot$ is element-wise product.

**Properties.** Different from traditional GNNs, GIB-Cat and GIB-Bern depend loosely on the graph structure since $A$ is only used to decide the potential neighbors for each node, and we perform message passing based on $Z_A$. This property renders our models extremely robust to structural perturbations/attacks where traditional GNNs are sensitive [15, 16]. Both our models also keep robustness to the feature perturbation that is similar to other IB-based DNN models [21, 26]. Moreover, the proposed models are invariant to node permutations as we may show that for any permutation matrix $\Pi \in \mathbb{R}^{n \times n}$, with permuting $A \to A_\Pi = \Pi A \Pi^T$, $X \to X_\Pi = \Pi X$, the obtained new node representations $Z_{X,\Pi}^{(L)}$ and $\Pi Z_X^{(L)}$ share the same distribution (proof in Appendix E). Permutation invariance is known to be important for structural representation learning [13].

---

**Algorithm 1: Framework of GIB-Cat and GIB-Bern**

**Input:** The dataset $\mathcal{D} = (X, A)$;
$\mathcal{T}$: An integral limitation to impose local dependence;
$k$: The number of neighbors to be sampled.
$\tau$: An element-wise nonlinear rectifier.
**Initialize:** $Z_X^{(0)} \leftarrow X$; For all $v \in V$, $t \in [\mathcal{T}]$, construct sets $V_{vt} \leftarrow \{u \in V | d(u, v) = t\}$;
Weights: $a \in \mathbb{R}^{\mathcal{T} \times 4f'}$, $W^{(1)} \in \mathbb{R}^{f \times 2f'}$, $W^{(l)} \in \mathbb{R}^{f' \times 2f'}$, for $l \in [2, L]$, $W_{\text{out}} \in \mathbb{R}^{f' \times K}$.
**Output:** $Z_X^{(L)}$, $\hat{Y}_v = \text{softmax}(Z_{X,v}^{(L)} W_{\text{out}})$
1. **For** layers $l = 1, ..., L$ and **For** $v \in V$, **do**:
2. $\quad$ $\tilde{Z}_{X,v}^{(l-1)} \leftarrow \tau(Z_{X,v}^{(l-1)}) W^{(l)}$
3. $\quad$ $Z_{A,v}^{(l)} \leftarrow$ **NeighborSample**$(Z_X^{l-1}, \mathcal{T}, V_{vt}, a)$
4. $\quad$ $\bar{Z}_{X,v}^{(l)} \leftarrow \sum_{u \in Z_{A,v}^{(l)}} \tilde{Z}_{X,v}^{(l-1)}$
5. $\quad$ $\mu_v^{(l)} \leftarrow \bar{Z}_{X,v}^{(l)}[0 : f']$
6. $\quad$ $\sigma_v^{2(l)} \leftarrow \text{softplus}(\bar{Z}_{X,v}^{(l)}[f' : 2f'])$
7. $\quad$ $Z_{X,v}^{(l)} \sim \text{Gaussian}(\mu_v^{(l)}, \sigma_v^{2(l)})$

---

| **Algorithm 2: NeighborSample (categorical)** | **Algorithm 3: NeighborSample (Bernoulli)** |
|---|---|
| **Input:** $Z_X^l, \mathcal{T}, V_{vt}, a$, as defined in Alg. 1; | **Input:** $Z_X^l, \mathcal{T}, V_{vt}, a$, as defined in Alg. 1; |
| **Output:** $Z_{A,v}^{(l+1)}$ | **Output:** $Z_{A,v}^{(l+1)}$ |
| 1.**For** $t \in [\mathcal{T}]$, **do**: | 1.**For** $t \in [\mathcal{T}]$, **do**: |
| 2. $\quad$ $\phi_{vt}^{(l)} \leftarrow \text{softmax}(\{(\tilde{Z}_{X,v}^{(l-1)} \oplus \tilde{Z}_{X,u}^{(l-1)}) a^T\}_{u \in V_{vt}})$ | 2. $\quad$ $\phi_{vt}^{(l)} \leftarrow \text{sigmoid}(\{(\tilde{Z}_{X,v}^{(l-1)} \oplus \tilde{Z}_{X,u}^{(l-1)}) a^T\}_{u \in V_{vt}})$ |
| 3. $Z_{A,v}^{(l+1)} \leftarrow \cup_{t=1}^{\mathcal{T}} \{u \in V_{vt} | u \overset{\text{iid}}{\sim} \text{Cat}(\phi_{vt}^{(l)}), k \text{ times}\}$ | 3. $Z_{A,v}^{(l+1)} \leftarrow \cup_{t=1}^{\mathcal{T}} \{u \in V_{vt} | u \overset{\text{iid}}{\sim} \text{Bernoulli}(\phi_{vt}^{(l)})\}$ |

**Objective for training.** To optimize the parameters of the model, we need to specify the bounds for $I(\mathcal{D}; Z_X^{(L)})$ as in Eq. (3) and $I(Y; Z_X^{(L)})$ as in Eq. (2), and further compute the bound of the GIB objective in Eq. (1). To characterize $\text{AIB}^{(l)}$ in Eq. (3), we assume $\mathbb{Q}(Z_A^{(l)})$ is a non-informative distribution [24, 25]. Specifically, we use the uniform distribution for the categorical version: $Z_A \sim \mathbb{Q}(Z_A)$, $Z_{A,v} = \cup_{t=1}^{\mathcal{T}} \{u \in V_{vt} | u \overset{\text{iid}}{\sim} \text{Cat}(\frac{1}{|V_{vt}|})\}$ and $Z_{A,v} \perp Z_{A,u}$ if $v \neq u$; and we also adopt a non-informative prior for the Bernoulli version: $Z_{A,v} = \cup_{t=1}^{\mathcal{T}} \{u \in V_{vt} | u \overset{\text{iid}}{\sim} \text{Bernoulli}(\alpha)\}$, where $\alpha \in (0, 1)$ is a hyperparameter. The difference is that, unlike the categorical distribution, we have an additional degree of freedom provided by $\alpha$. After the model computes $\phi_{vt}^{(l)}$ according to Step 4, we

get an empirical estimation of AIB$^{(l)}$:

$$\widehat{\text{AIB}}^{(l)} = \mathbb{E}_{\mathbb{P}(Z_A^{(l)}|A, Z_X^{(l-1)})} \left[ \log \frac{\mathbb{P}(Z_A^{(l)}|A, Z_X^{(l-1)})}{\mathbb{Q}(Z_A^{(l)})} \right],$$

which is instantiated as follows for the two versions,

$$\widehat{\text{AIB}_\text{C}}^{(l)} = \sum_{v \in V, t \in [\mathcal{T}]} \text{KL}(\text{Cat}(\phi_{vt}^{(l)}) || \text{Cat}(\frac{\mathbf{1}}{|V_{vt}|}))$$

$$\widehat{\text{AIB}_\text{B}}^{(l)} = \sum_{v \in V, t \in [\mathcal{T}]} \text{KL}(\text{Bernoulli}(\phi_{vt}^{(l)}) || \text{Bernoulli}(\alpha))$$

To estimate XIB$^{(l)}$, we set $\mathbb{Q}(Z_X^{(l)})$ as a mixture of Gaussians with learnable parameters [27]. Specifically, for any node $v$, $Z_X \sim \mathbb{Q}(Z_X)$, we set $Z_{X,v} \sim \sum_{i=1}^m w_i \text{Gaussian}(\mu_{0,i}, \sigma_{0,i}^2)$ where $w_i, \mu_{0,i}, \sigma_{0,i}$ are learnable parameters shared by all nodes and $Z_{X,v} \perp Z_{X,u}$ if $v \neq u$. We estimate XIB$^{(l)}$ by using the sampled $Z_X^{(l)}$:

$$\widehat{\text{XIB}}^{(l)} = \log \frac{\mathbb{P}(Z_X^{(l)}|Z_X^{(l-1)}, Z_A^{(l)})}{\mathbb{Q}(Z_X^{(l)})} = \sum_{v \in V} \left[ \log \Phi(Z_{X,v}^{(l)}; \mu_v, \sigma_v^2) - \log(\sum_{i=1}^m w_i \Phi(Z_{X,v}^{(l)}; \mu_{0,i}, \sigma_{0,i}^2)) \right].$$

Therefore, in practice, we may select proper sets of indices $S_X$, $S_A$ that satisfy the condition in Proposition 3.2 and use substitution

$$I(\mathcal{D}; Z_X^{(L)}) \rightarrow \sum_{l \in S_A} \widehat{\text{AIB}}^{(l)} + \sum_{l \in S_X} \widehat{\text{XIB}}^{(l)} \tag{5}$$

To characterize Eq. (2), we may simply set $\mathbb{Q}_2(Y) = \mathbb{P}(Y)$ and $\mathbb{Q}_1(Y_v|Z_{X,v}^{(L)}) = \text{Cat}(Z_{X,v}^{(L)} W_\text{out})$. Then, the RHS of Eq. (2) reduces to the cross-entropy loss by ignoring constants, *i.e.*,

$$I(Y; Z_X^{(L)}) \rightarrow - \sum_{v \in V} \text{Cross-Entropy}(Z_{X,v}^{(L)} W_\text{out}; Y_v) \tag{6}$$

Other choices of $\mathbb{Q}_2(Y)$ may also be adopted and yield the contrastive loss [22,28] (Appendix D). However, in our case, we use the simplest setting to illustrate the benefit of the GIB principle. Plugging Eq. (5) and Eq. (6) into Eq. (1), we obtain the objective to train our models.

**Other Formalizations of the GIB Principle.** There are also other alternative formalizations of the GIB principle, especially when modeling $\mathbb{P}(Z_A^{(l)}|Z_X^{(l-1)}, A)$. Generally speaking, any node-pair representations, such as messages over edges in MPNN [29], can be leveraged to sample structures. Applying the GIB principle to other architectures is a promising direction for future investigation.

## 4 Related Work

GNNs learn node-level representations through message passing and aggregation from neighbors [1,3,29–31]. Several previous works further incorporate the attention mechanism to adaptively learn the correlation between a node and its neighbor [5,32]. Recent literature shows that representations learned by GNNs are far from robust and can be easily attacked by malicious manipulation on either features or structure [15,16]. Accordingly, several defense models are proposed to increase the robustness by injecting random noise in the representations [33], removing suspicious and uninformative edges [34], low-rank approximation of the adjacency matrix [35], additional hinge loss for certified robustness [36]. In contrast, even though not specifically designed against adversarial attacks, our model learns robust representations via the GIB principle that naturally defend against attacks. Moreover, none of those defense models has theoretical foundations except [36] that uses tools of robust optimization instead of information theory.

Recently several works have applied contrastive loss [28] as a regularizer for GNNs. The idea is to increase the score for positive samples while decrease the score for negative samples. This

can be further formulated as a mutual information maximization term that aims to maximize the mutual information between representations of nodes and their neighbor patches [37], between representations of sub-structures and the hidden feature vectors [38], between representations of graphs and their sub-structures [39]. In contrast, our model focuses on the compression of node features and graph structure while at the same time improves prediction, which is orthogonal to these previous works on unsupervised representation learning with information maximization.

Another line of related work is representation learning with the IB principle. DVIB [21] first applies IB [18] to deep neural networks, and shows increased robustness of learned representations. Other methods apply IB to various domains [40, 41]. The difference is that we develop information-theoretic modeling of feature, structure and their fusion on graph-structured data. Furthermore, several works on GNNs [37–39] leverage information maximization [42] for unsupervised learning. However, we focus on learning robust representations by *controlling* the information in supervised learning setting.

## 5 Experiments

The goal of our experiments is to test whether GNNs trained with the GIB objective are more robust and reliable. Specifically, we consider the following two questions: (1) Boosted by GIB, does GIB-Cat and GIB-Bern learn more robust representations than GAT to defend against attacks? (2) How does each component of GIB contribute to such robustness, especially, to controlling the information from one of the two sides — the structure and node features?

We compare GIB-Cat and GIB-Bern with baselines including GCN [3] and GAT [5], the most relevant baseline as GIB-Cat and GIB-Bern are to impose the GIB principle over GAT. In addition, we consider two state-of-the-art graph defense models specifically designed against adversarial attacks: GCNJaccard [34] that pre-processes the graph by deleting the edges between nodes with low feature similarity, and Robust GCN (RGCN) [33] that uses Gaussian reparameterization for node features and variance-based attention. Note that RGCN essentially includes the term XIB (Eq. (3)) to control the information of node features while it does not have the term AIB (Eq. (3)) to control the structural information. For GCNJaccard and RGCN, we perform extensive hyperparameter search as detailed in Appendix G.3. For GIB-Cat and GIB-Bern, we keep the same architectural component as GAT, and for the additional hyperparameters $k$ and $\mathcal{T}$ (Algorithm 1, 2 and 3), we search $k \in \{2, 3\}$ and $\mathcal{T} \in \{1, 2\}$ for each experimental setting and report the better performance. Please see Appendix G for more details.

We use three citation benchmark datasets: Cora, Pubmed and Citeseer [43], in our evaluation. In all experiments, we follow the standard *transductive* node classification setting and standard train-validation-test split as GAT [5]. The summary statistics of the datasets and their splitting are shown in Table 4 in Appendix F. For all experiments, we perform the experiments over 5 random initializations and report average performance. We always use F1-micro as the validating metric to train our model.

### 5.1 Robustness Against Adversarial Attacks

In this experiment, we compare the robustness of different models against adversarial attacks. We use Nettack [15], a strong targeted attack technique on graphs that attacks a target node by flipping the edge or node features. We evaluate the models on both evasive and poisoning settings, *i.e.* the attack happens after or before the model is trained, respectively. We follow the setting of Nettack [15]: for each dataset, select (i) 10 nodes with highest margin of classification, *i.e.* they are clearly correctly classified, (ii) 10 nodes with lowest margin but still correctly classified and (iii) 20 more nodes randomly, where for each target node, we train a different model for evaluation. We report the classification accuracy of these 40 targeted nodes. We enumerate the number of perturbations from 1 to 4, where each perturbation denotes a flipping of a node feature or an addition or deletion of an edge. Since Nettack can only operate on Boolean features, we binarize the node features before training.

Table 1 shows the results. We see that compared with GAT, GIB-Cat improves the classification accuracy by an average of 8.9% and 14.4% in Cora and Pubmed, respectively, and GIB-Bern improves the classification accuracy by an average of 8.4% and 14.6% in Cora and Pubmed, respectively, which demonstrates the effectiveness of the GIB principle to improve the robustness of GNNs. Remarkably, when the number of perturbation is 1, GIB-Cat and GIB-Bern boost accuracy over GAT (as well as other models) by 31.3% and 34.0% in Pubmed, respectively. GIB-Cat also outperforms GCNJaccard

Table 1: Average classification accuracy (%) for the targeted nodes under direct attack. Each number is the average accuracy for the 40 targeted nodes for 5 random initialization of the experiments. Bold font denotes top two models.

| | Model | Clean (%) | Evasive (%) | | | | Poisoning (%) | | | |
|---|---|---|---|---|---|---|---|---|---|---|
| | | | 1 | 2 | 3 | 4 | 1 | 2 | 3 | 4 |
| Cora | GCN | **80.0**±7.87 | 51.5±4.87 | 38.0±6.22 | 31.0±2.24 | 26.0±3.79 | 47.5±7.07 | 39.5±2.74 | 30.0±5.00 | 26.5±3.79 |
| | GCNJaccard | 75.0±5.00 | 48.5±6.75 | 36.0±6.51 | 32.0±3.25 | 30.0±3.95 | 47.0±7.37 | 38.0±6.22 | 33.5±3.79 | 28.5±3.79 |
| | RGCN | **80.0**±4.67 | 49.5±6.47 | 36.0±5.18 | 30.5±3.25 | 25.5±2.09 | 46.5±5.75 | 35.5±3.70 | 29.0±3.79 | 25.5±2.73 |
| | GAT | 77.8±3.97 | 48.0±8.73 | 39.5±5.70 | 36.5±5.48 | 32.5±5.30 | 50.5±5.70 | 38.0±5.97 | 33.5±2.85 | 26.0±3.79 |
| | **GIB-Cat** | 77.6±2.84 | **63.0**±4.81 | **52.5**±3.54 | **44.5**±5.70 | **36.5**±6.75 | **60.0**±6.37 | **50.0**±2.50 | **39.5**±5.42 | **30.0**±3.95 |
| | **GIB-Bern** | 78.4±4.07 | **64.0**±5.18 | **51.5**±4.54 | **43.0**±3.26 | **37.5**±3.95 | **61.5**±4.18 | **46.0**±4.18 | **36.5**±4.18 | **31.5**±2.85 |
| Pubmed | GCN | 82.6±6.98 | 39.5±4.81 | 32.0±4.81 | 31.0±5.76 | 31.0±5.76 | 36.0±4.18 | 32.5±6.37 | 31.0±5.76 | 28.5±5.18 |
| | GCNJaccard | 82.0±7.15 | 37.5±5.30 | 31.5±5.18 | 30.0±3.95 | 30.0±3.95 | 36.0±3.79 | 32.5±4.67 | 31.0±4.87 | 28.5±4.18 |
| | RGCN | 79.0±5.18 | 39.5±5.70 | 33.0±4.80 | 31.5±4.18 | 30.0±5.00 | 38.5±4.18 | 31.5±2.85 | 29.5±3.70 | 27.0±3.70 |
| | GAT | 78.6±6.70 | 41.0±8.40 | 33.5±4.18 | 30.5±4.47 | 31.0±4.18 | 39.5±3.26 | 31.0±4.18 | 30.0±3.06 | 25.5±5.97 |
| | **GIB-Cat** | **85.1**±6.90 | **72.0**±3.26 | **51.0**±5.18 | **37.5**±5.30 | **31.5**±4.18 | **71.0**±4.87 | **48.0**±3.26 | **37.5**±1.77 | **28.5**±2.24 |
| | **GIB-Bern** | **86.2**±6.54 | **76.0**±3.79 | **50.5**±4.11 | **37.5**±3.06 | **31.5**±1.37 | **72.5**±4.68 | **48.0**±2.74 | **36.0**±2.85 | 26.5±2.85 |
| Citeseer | GCN | 71.8±6.94 | 42.5±7.07 | 27.5±6.37 | 18.0±3.26 | 15.0±2.50 | 29.0±7.20 | 20.5±1.12 | 17.5±1.77 | 13.0±2.09 |
| | GCNJaccard | 72.5±9.35 | 41.0±6.75 | 32.5±3.95 | 20.5±3.70 | 13.0±1.11 | **42.5**±5.86 | **30.5**±5.12 | 17.5±1.76 | **14.0**±1.36 |
| | RGCN | **73.5**±8.40 | 41.5±7.42 | 24.5±6.47 | 18.5±6.52 | 13.0±1.11 | 31.0±5.48 | 19.5±2.09 | 13.5±2.85 | 5.00±1.77 |
| | GAT | 72.3±8.38 | **49.0**±9.12 | 33.0±5.97 | 22.0±4.81 | 18.0±3.26 | **38.0**±5.12 | **23.5**±4.87 | 16.5±4.54 | 12.0±2.09 |
| | **GIB-Cat** | 68.6±4.90 | **51.0**±4.54 | **39.0**±4.18 | **32.0**±4.81 | **26.5**±4.54 | 30.0±9.19 | 14.0±5.76 | 9.50±3.26 | 6.50±2.24 |
| | **GIB-Bern** | 71.8±5.03 | **49.0**±7.42 | **37.5**±7.71 | **32.5**±4.68 | **23.5**±7.42 | 35.0±6.37 | 19.5±4.81 | 11.5±3.79 | 6.00±2.85 |

Table 2: Average classification accuracy (%) for the ablations of GIB-Cat and GIB-Bern on Cora dataset.

| Model | Clean (%) | Evasive (%) | | | | Poisoning (%) | | | |
|---|---|---|---|---|---|---|---|---|---|
| | | 1 | 2 | 3 | 4 | 1 | 2 | 3 | 4 |
| **XIB** | 76.3±2.90 | 57.0±5.42 | 47.5±7.50 | 39.5±6.94 | 33.0±3.71 | 54.5±2.09 | 41.0±3.79 | 36.0±5.18 | 31.0±4.54 |
| **AIB-Cat** | 78.7±4.95 | 62.5±5.86 | 51.5±5.18 | 43.0±3.26 | 36.0±3.35 | 60.5±3.26 | 47.5±5.00 | 36.0±3.35 | 31.5±6.27 |
| **AIB-Bern** | 79.9±3.78 | 64.0±4.50 | 51.5±6.50 | 42.0±5.40 | 37.0±5.70 | 58.5±3.80 | 46.0±4.50 | 39.0±4.20 | 30.0±3.10 |
| **GIB-Cat** | 77.6±2.84 | 63.0±4.81 | 52.5±3.54 | 44.5±5.70 | 36.5±6.75 | 60.0±6.37 | 50.0±2.50 | 39.5±5.42 | 30.0±3.95 |
| **GIB-Bern** | 78.4±4.07 | 64.0±5.18 | 51.5±4.54 | 43.0±3.26 | 37.5±3.95 | 61.5±4.18 | 46.0±4.18 | 36.5±4.18 | 31.5±2.85 |

and RGCN by an average of 10.3% and 12.3% on Cora (For GIB-Bern, it is 9.8% and 11.7%), and by an average of 15.0% and 14.6% on Pubmed (For GIB-Bern, it is 15.2% and 14.8%), although GIB-Cat and GIB-Bern are not intentionally designed to defend attacks. For Citeseer, GIB-Cat and GIB-Bern's performance are worse than GCNJaccard in the poisoning setting. This is because Citeseer has much more nodes with very few degrees, even fewer than the number of specified perturbations, as shown in Table 13 in Appendix J. In this case, the most effective attack is to connect the target node to a node from a different class with very different features, which exactly matches the assumption used by GCNJaccard [34]. GCNJaccard proceeds to delete edges with dissimilar node features, resulting in the best performance in Citeseer. However, GIB does not depend on such a restrictive assumption. More detailed analysis is at Appendix J.

**Ablation study.** To see how different components of GIB contribute to the performance, we perform ablation study on Cora, as shown in Table 2. Here, we use AIB-Cat and AIB-Bern to denote the models that only sample structures with $\widehat{\text{AIB}}$ (Eq. (5)) in the objective (whose NeighborSample() function is identical to that of GIB-Cat and GIB-Bern, respectively), and use XIB to denote the model that only samples node representations with $\widehat{\text{XIB}}$ (Eq. (5)) in the objective. We see that the AIB (structure) contributes significantly to the improvement of GIB-Cat and GIB-Bern, and on average, AIB-Cat (AIB-Bern) only underperforms GIB-Cat (GIB-Bern) by 0.9% (0.4%). The performance gain is due to the attacking style of Nettack, as the most effective attack is typically via structural perturbation [15], as is also confirmed in Appendix J. Therefore, next we further investigate the case that only perturbation on node features is available.

## 5.2 Only Feature Attacks

To further check the effectiveness of IB for node features, we inject random perturbation into the node features. Specifically, after the models are trained, we add independent Gaussian noise to each dimension of the node features for all nodes with increasing amplitude. Specifically, we use the mean of the maximum value of each node's feature as the reference amplitude $r$, and for each feature dimension of each node we add Gaussian noise $\lambda \cdot r \cdot \epsilon$, where $\epsilon \sim N(0, 1)$, and $\lambda$ is the feature noise ratio. We test the models' performance with $\lambda \in \{0.5, 1, 1.5\}$. Table 3 shows the results. We

Table 3: Classification F1-micro (%) for the trained models with increasing additive feature noise. Bold font denotes top 2 models.

| Dataset | Model | Feature noise ratio ($\lambda$) | | |
|---|---|---|---|---|
| | | 0.5 | 1 | 1.5 |
| Cora | GCN | $64.0_{\pm2.05}$ | $41.3_{\pm2.05}$ | $31.4_{\pm2.81}$ |
| | GCNJaccard | $61.1_{\pm2.18}$ | $41.2_{\pm2.28}$ | $31.8_{\pm2.63}$ |
| | RGCN | $57.7_{\pm2.27}$ | $39.1_{\pm1.58}$ | $29.6_{\pm2.47}$ |
| | GAT | $62.5_{\pm1.97}$ | $41.7_{\pm2.32}$ | $29.8_{\pm2.98}$ |
| | AIB-Cat | $67.9_{\pm2.65}$ | $\mathbf{49.6}_{\pm5.35}$ | $\mathbf{38.4}_{\pm5.06}$ |
| | AIB-Bern | $\mathbf{68.8}_{\pm1.85}$ | $49.0_{\pm2.87}$ | $37.1_{\pm4.47}$ |
| | **GIB-Cat** | $67.1_{\pm2.21}$ | $49.1_{\pm3.67}$ | $37.5_{\pm4.76}$ |
| | **GIB-Bern** | $\mathbf{69.0}_{\pm1.91}$ | $\mathbf{51.3}_{\pm2.62}$ | $\mathbf{38.9}_{\pm3.38}$ |
| Pubmed | GCN | $61.3_{\pm1.52}$ | $50.2_{\pm2.08}$ | $44.3_{\pm1.43}$ |
| | GCNJaccard | $62.7_{\pm1.25}$ | $51.9_{\pm1.53}$ | $45.1_{\pm2.04}$ |
| | RGCN | $58.4_{\pm1.74}$ | $49.0_{\pm1.65}$ | $43.9_{\pm1.29}$ |
| | GAT | $62.7_{\pm1.68}$ | $50.2_{\pm2.35}$ | $43.7_{\pm2.43}$ |
| | AIB-Cat | $64.5_{\pm2.13}$ | $50.9_{\pm3.83}$ | $43.0_{\pm3.73}$ |
| | AIB-Bern | $61.1_{\pm2.70}$ | $47.8_{\pm3.65}$ | $42.0_{\pm4.21}$ |
| | **GIB-Cat** | $\mathbf{67.1}_{\pm4.33}$ | $\mathbf{57.2}_{\pm5.27}$ | $\mathbf{51.5}_{\pm4.84}$ |
| | **GIB-Bern** | $\mathbf{64.9}_{\pm2.52}$ | $\mathbf{54.7}_{\pm1.83}$ | $\mathbf{48.2}_{\pm2.10}$ |
| Citeseer | GCN | $\mathbf{55.9}_{\pm1.33}$ | $40.6_{\pm1.83}$ | $32.8_{\pm2.19}$ |
| | GCNJaccard | $\mathbf{56.8}_{\pm1.49}$ | $41.3_{\pm1.81}$ | $33.1_{\pm2.27}$ |
| | RGCN | $51.4_{\pm2.00}$ | $36.5_{\pm2.38}$ | $29.5_{\pm2.17}$ |
| | GAT | $55.8_{\pm1.43}$ | $40.8_{\pm1.77}$ | $33.8_{\pm1.93}$ |
| | AIB-Cat | $55.1_{\pm1.26}$ | $43.1_{\pm2.46}$ | $35.6_{\pm3.19}$ |
| | **AIB-Bern** | $55.8_{\pm2.01}$ | $\mathbf{43.3}_{\pm1.67}$ | $\mathbf{36.3}_{\pm2.47}$ |
| | GIB-Cat | $54.9_{\pm1.39}$ | $42.0_{\pm1.92}$ | $34.8_{\pm1.75}$ |
| | **GIB-Bern** | $54.4_{\pm5.98}$ | $\mathbf{50.3}_{\pm4.33}$ | $\mathbf{46.1}_{\pm2.47}$ |

see across different feature noise ratios, both GIB-Cat and GIB-Bern consistently outperforms other models without IB, especially when the feature noise ratio is large ($\lambda = 1.5$), and the AIB models with only structure IB performs slightly worse or equivalent to the GIB models. This shows that GIB makes the model more robust when the feature attack becomes the main source of perturbation.

## 6   Conclusion and Discussion

In this work, we have introduced Graph Information Bottleneck (GIB), an information-theoretic principle for learning representations that capture minimal sufficient information from graph-structured data. We have also demonstrated the efficacy of GIB by evaluating the robustness of the GAT model trained under the GIB principle on adversarial attacks. Our general framework leaves many interesting questions for future investigation. For example, are there any other better instantiations of GIB, especially in capturing discrete structural information? If incorporated with a node for global aggregation, can GIB break the limitation of the local-dependence assumption? May GIB be applied to other graph-related tasks including link prediction and graph classification?

## Broader Impact

**Who may benefit from this research:** Graphs have been used to represent a vast amount of real-world data from social science [44], biology [45], geographical mapping [46], finances [47] and recommender systems [48], because of their flexibility in modeling both the relation among the data (structures) and the content of the data (features). Graph neural networks (GNN), naturally entangle both aspects of the data in the most expressive way, have attracted unprecedented attention from both academia and industry across a wide range of disciplines. However, GNNs share a common issue with other techniques based on neural networks. They are very sensitive to noise of data and are fragile to model attacks. This drawback yields the potential safety problems to deploy GNNs in the practical systems or use them to process data in those disciplines that heavily emphasize unbiased analysis. The Graph Information Bottleneck (GIB) principle proposed in this work paves a principled way to alleviate the above problem by increasing the robustness of GNN models. Our work further releases the worries about the usage of GNN techniques in practical systems, such as recommender systems, social media, or to analyze data for other disciplines, including physics, biology, social science. Ultimately, our work increases the interaction between AI, machine learning techniques and other aspects of our society, and could achieve far-reaching impact.

**Who may be put at disadvantage from this research:** Not applicable.

**What are the consequences of failure of the system:** Not applicable.

**Does the task/method leverage biases in the data:** The proposed GIB principle and the GIB-GAT model as an instantiation of GIB leverage the node features and structural information which in general are not believed to include undesirable biases. The datasets to evaluate our approaches are among the most widely-used benchmarks, which in general are not believed to include undesirable biases as well.

## Acknowledgments and Disclosure of Funding

We thank the anonymous reviewers for providing feedback on our manuscript. Hongyu Ren is supported by the Masason Foundation Fellowship. Jure Leskovec is a Chan Zuckerberg Biohub investigator. We also gratefully acknowledge the support of DARPA under Nos. FA865018C7880 (ASED), N660011924033 (MCS); ARO under Nos. W911NF-16-1-0342 (MURI), W911NF-16-1-0171 (DURIP); NSF under Nos. OAC-1835598 (CINES), OAC-1934578 (HDR), CCF-1918940 (Expeditions), IIS-2030477 (RAPID); Stanford Data Science Initiative, Wu Tsai Neurosciences Institute, Chan Zuckerberg Biohub, Amazon, Boeing, JPMorgan Chase, Docomo, Hitachi, JD.com, KDDI, NVIDIA, Dell.

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
