[Supplementary Material]

# Appendix

## A  Preliminaries for Information Bottleneck

Here we briefly review the Information Bottleneck (IB) principle and its application to representation learning.

Given the input data $\mathcal{D}$ and target $Y$, and an stochastic encoding $Z$ of $\mathcal{D}$ by $\mathbb{P}(Z|\mathcal{D})$ that satisfies the Markov chain $Z - \mathcal{D} - Y$, IB has the following objective:

$$\min_{\mathbb{P}(Z|\mathcal{D})} \text{IB}_\beta(\mathcal{D}, Y; Z) := [-I(Y; Z) + \beta I(\mathcal{D}; Z)] \tag{7}$$

It also has an equivalent form:

$$\max_{\mathbb{P}(Z|\mathcal{D}):I(\mathcal{D};Z)\leq I_c} I(Y; Z) \tag{8}$$

Intuitively, Eq. (7) or (8) encourages the representation $Z$ to maximally capture the information in $Y$, while controlling the complexity of the representation in terms of $I(\mathcal{D}; Z)$. When increasing $\beta$ from 0 to some large value, we are essentially using a straight line with slope $\beta$ to sweep out the Pareto frontier of $I(Y; Z)$ vs. $I(X; Z)$ as given by Eq. (8).

Figure 3: Information diagram for the Information Bottleneck (IB). Also plotted are the minimal sufficient information as covered by $I(\mathcal{D}; Y)$ and overfitting part that occupies parts of $H(\mathcal{D}|Y)$.

Using the information diagram (Fig. 3), where we represent the information of $\mathcal{D}, Y$ as circles and their shared part as the overlapping region of the circles, then IB encourages $Z$ to cover as much of the $I(\mathcal{D}; Y)$ as possible, and cover as little of $H(\mathcal{D}|Y)$ (the irrelevant information part) as possible. An optimal representation is defined as the minimal sufficient representation [49] that only covers $I(\mathcal{D}; Y)$. In practice, due to the expressiveness of the models and different choices of $\beta$ in Eq. (7), this optimal information can hardly be reached, and may only be approached. It is an interesting future direction to study that when sweeping $\beta$, how near it is to the optimal representation on the diagram of $I(Y; Z)$ vs. $I(X; Z)$.

## B  Proof for Proposition 3.1

We restate Proposition 3.2: For any PDFs $\mathbb{Q}_1(Y_v|Z_{X,v}^{(L)})$ for $v \in V$ and $\mathbb{Q}_2(Y)$, we have

$$I(Y; Z_X^{(L)}) \geq 1 + \mathbb{E}\left[\log \frac{\prod_{v\in V} \mathbb{Q}_1(Y_v|Z_{X,v}^{(L)})}{\mathbb{Q}_2(Y)}\right] + \mathbb{E}_{\mathbb{P}(Y)\mathbb{P}(Z_X^{(L)})}\left[\frac{\prod_{v\in V} \mathbb{Q}_1(Y_v|Z_{X,v}^{(L)})}{\mathbb{Q}_2(Y)}\right] \tag{9}$$

*Proof.* We use the Nguyen, Wainright & Jordan's bound $I_{\text{NWJ}}$ [22, 23]:

**Lemma B.1.** [22, 23] For any two random variables $X_1, X_2$ and any function $g : g(X_1, X_2) \in \mathbb{R}$, we have

$$I(X_1, X_2) \geq \mathbb{E}[g(X_1, X_2)] - \mathbb{E}_{\mathbb{P}(X_1)\mathbb{P}(X_2)}[\exp(g(X_1, X_2) - 1)].$$

We use the above lemma to $(Y, Z_X^{(L)})$ and plug in $g(Y, Z_X^{(L)}) = 1 + \log \frac{\prod_{v\in V} \mathbb{Q}_1(Y_v|Z_{X,v}^{(L)})}{\mathbb{Q}_2(Y)}$.  $\square$

# C Proof for Proposition 3.2

We restate Proposition 3.2: For any groups of indices $S_X, S_A \subset [L]$ such that $\mathcal{D} \perp Z_X^{(L)} | \{Z_X^{(l)}\}_{l \in S_X} \cup \{Z_A^{(l)}\}_{l \in S_A}$, and for any probabilistic distributions $\mathbb{Q}(Z_X^{(l)}), l \in S_X$, and $\mathbb{Q}(Z_A^{(l)})$, $l \in S_A$,

$$I(\mathcal{D}; Z_X^{(L)}) \leq I(\mathcal{D}; \{Z_X^{(l)}\}_{l \in S_X} \cup \{Z_A^{(l)}\}_{l \in S_A}) \leq \sum_{l \in S_X} \mathrm{XIB}^{(l)} + \sum_{l \in S_A} \mathrm{AIB}^{(l)}, \text{ where} \quad (10)$$

$$\mathrm{AIB}^{(l)} = \mathbb{E}\left[\log \frac{\mathbb{P}(Z_A^{(l)} | A, Z_X^{(l-1)})}{\mathbb{Q}(Z_A^{(l)})}\right], \mathrm{XIB}^{(l)} = \mathbb{E}\left[\log \frac{\mathbb{P}(Z_X^{(l)} | Z_X^{(l-1)}, Z_A^{(l)})}{\mathbb{Q}(Z_X^{(l)})}\right], \quad (11)$$

*Proof.* The first inequality $I(\mathcal{D}; Z_X^{(L)}) \leq I(\mathcal{D}; \{Z_X^{(l)}\}_{l \in S_X} \cup \{Z_A^{(l)}\}_{l \in S_A})$ directly results from the data processing inequality [17] and the Markov property $\mathcal{D} \perp Z_X^{(L)} | \{Z_X^{(l)}\}_{l \in S_X} \cup \{Z_A^{(l)}\}_{l \in S_A}$.

To prove the second inequality, we define an order "$\prec$" of random variables in $\{Z_X^{(l)}\}_{l \in S_X} \cup \{Z_A^{(l)}\}_{l \in S_A}$ such that 1) for two different integers $l, l', Z_X^{(l)}, Z_A^{(l)} \prec Z_X^{(l')}, Z_A^{(l')}$; 2) For one integer $l$, $Z_A^{(l)} \prec Z_X^{(l)}$. Based on the order, define a sequence of sets

$$H_A^{(l)} = \{Z_X^{(l_1)}, Z_A^{(l_2)} | l_1 < l, l_2 < l, l_1 \in S_X, l_2 \in S_A\},$$
$$H_X^{(l)} = \{Z_X^{(l_1)}, Z_A^{(l_2)} | l_1 < l, l_2 \leq l, l_1 \in S_X, l_2 \in S_A\}.$$

We may decompose $I(\mathcal{D}; \{Z_X^{(l)}\}_{l \in S_X} \cup \{Z_A^{(l)}\}_{l \in S_A})$ with respect to this order

$$I(\mathcal{D}; \{Z_X^{(l)}\}_{l \in S_X} \cup \{Z_A^{(l)}\}_{l \in S_A}) = \sum_{l \in S_A} I(\mathcal{D}; Z_A^{(l)} | H_A^{(l)}) + \sum_{l \in S_X} I(\mathcal{D}; Z_X^{(l)} | H_X^{(l)}).$$

Next, we bound each term in the RHS

$$\begin{aligned}
I(\mathcal{D}; Z_A^{(l)} | H_A^{(l)}) &\overset{1)}{\leq} I(\mathcal{D}, Z_X^{(l-1)}; Z_A^{(l)} | H_A^{(l)}) \\
&\overset{2)}{=} I(Z_X^{(l-1)}, A; Z_A^{(l)} | H_A^{(l)}) + I(X; Z_A^{(l)} | H_A^{(l)}, A, Z_X^{(l-1)}) \\
&\overset{3)}{=} I(Z_X^{(l-1)}, A; Z_A^{(l)} | H_A^{(l)}) + 0 \\
&\overset{4)}{\leq} I(Z_X^{(l-1)}, A; Z_A^{(l)}) \\
&\overset{5)}{=} \mathrm{AIB}^{(l)} - \mathrm{KL}(\mathbb{P}(Z_A^{(l)}) || \mathbb{Q}(Z_A^{(l)})) \leq \mathrm{AIB}^{(l)}
\end{aligned}$$

$$\begin{aligned}
I(\mathcal{D}; Z_X^{(l)} | H_X^{(l)}) &\overset{1)}{\leq} I(\mathcal{D}, Z_X^{(l-1)}, Z_A^{(l)}; Z_X^{(l)} | H_X^{(l)}) \\
&\overset{2)}{=} I(Z_X^{(l-1)}, Z_A^{(l)}; Z_X^{(l)} | H_X^{(l)}) + I(\mathcal{D}; Z_X^{(l)} | H_X^{(l)}, Z_X^{(l-1)}, Z_A^{(l)}) \\
&\overset{3)}{=} I(Z_X^{(l-1)}, Z_A^{(l)}; Z_X^{(l)} | H_X^{(l)}) + 0 \\
&\overset{4)}{\leq} I(Z_X^{(l-1)}, Z_A^{(l)}; Z_X^{(l)}) \\
&\overset{5)}{=} \mathrm{XIB}^{(l)} - \mathrm{KL}(\mathbb{P}(Z_X^{(l)}) || \mathbb{Q}(Z_X^{(l)})) \leq \mathrm{XIB}^{(l)}
\end{aligned}$$

where 1), 2) use the basic properties of mutual information, 3) uses $X \perp Z_A^{(l)} | \{A, Z_X^{(l-1)}\}$ and $\mathcal{D} \perp Z_X^{(l)} | \{Z_A^{(l-1)}, Z_X^{(l-1)}\}$, 4) uses $H_A^{(l)} \perp Z_A^{(l)} | \{Z_X^{(l-1)}, A\}$ and $H_X^{(l)} \perp Z_X^{(l)} | \{Z_X^{(l-1)}, Z_A^{(l)}\}$ and 5) uses the definitions of $\mathrm{AIB}^{(l)}$ and $\mathrm{XIB}^{(l)}$. $\qquad\square$

# D The Contrastive Loss Derived from the Variational Bound Eq. (2)

To characterize Eq. (2), We may also use a contrastive loss [22, 28] which empirically may sometimes improve the robustness of the model. Concretely, we keep $\mathbb{Q}_1(Y_v | Z_{X,v}^{(L)})$ as the

same as that to derive Eq. (6), i.e., $\mathbb{Q}_1(Y_v|Z_{X,v}^{(L)}) = \mathrm{Cat}(Z_{X,v}^{(L)}W_{\mathrm{out}})$ and set $\mathbb{Q}_2(Y) = \mathbb{E}_{\mathbb{P}(Z_X^{(L)})\mathbb{P}(Z_X^{'(L)})}[\prod_{v\in V}\frac{1}{2}(\mathbb{Q}_1(Y_v|Z_{X,v}^{(L)}) + \mathbb{Q}_1(Y_v|Z_{X,v}^{'(L)}))]$. Here, $\mathbb{P}(Z_X^{'(L)})$ refers to the distribution of the last-layer node representation after we replace $A$ with a random graph structure $A' \in \mathbb{R}^{n\times n}$ where $A'$ is uniformly sampled with the constraint that $A'$ has the same number of edges as $A$. When using this contrastive loss, we simply use the estimation of $\mathbb{Q}_2(Y)$ based on the sampled $Z_{X,v}^{(L)}$ and $Z_{X,v}^{'(L)}$. Moreover, the last term of Eq. (2) is empirically closed to 1 and thus we ignore it and other constants in Eq. (2). Overall, we have the substitution for the contrastive loss,

$$I(Y;Z_X^{(L)}) \rightarrow \sum_{v\in V}\left[\log(h(Y_v;Z_{X,v}^{(L)})) - \log(h(Y_v;Z_{X,v}^{(L)}) + h(Y_v;Z_{X,v}^{'(L)}))\right], \qquad (12)$$

where $h(Y_v;Z_{X,v}) = \frac{\exp(Z_{X,v}W_{\mathrm{out}}[Y_v])}{\sum_{i=1}^{K}\exp(Z_{X,v}W_{\mathrm{out}}[i])}$.

# E   Permutation Invariance of GIB-Cat and GIB-Bern

Let $\Pi \in \mathbb{R}^{n\times n}$ denote a permutation matrix where each row and each column contains exactly one single 1 and the rest components are all 0's. For any variable in GIB-Cat or GIB-Bern, we use subscript $\Pi$ to denote the corresponding obtained variable after we permutate the node indices of the input data, i.e., $\mathcal{D} = (X, A) \rightarrow \Pi(\mathcal{D}) = (\Pi X, \Pi A\Pi^T)$. For example, $Z_{X,\Pi}^{(l)}$ denotes the node representations after $l$ layers of GIB-Cat or GIB-Bern based on the input data $\Pi(\mathcal{D})$. Moreover, the matrix $\Pi$ also defines a bijective mapping $\pi : V \rightarrow V$, where $\pi(v) = u$ iff $\Pi_{uv} = 1$. We also use "$\overset{d}{=}$" to denote that two random variables share the same distribution.

Now, we formally restate the permutation invariant property of GIB-Cat and GIB-Bern: Suppose $\Pi \in \mathbb{R}^{n\times n}$ is any permutation matrix, if the input graph-structured data becomes $\Pi(\mathcal{D}) = (\Pi X, \Pi A\Pi^T)$, the corresponding node representations output by GIB-Cat or GIB-Bern satisfy $Z_{X,\Pi}^{(L)} \overset{d}{=} \Pi Z_X^{(L)}$ where $Z_X^{(L)}$ is the output node representations based on the original input data $\mathcal{D} = (X, A)$.

*Proof.* We use induction to prove this result. Specifically, we only need to show that for a certain $l \in [L]$, if node representations $Z_{X,\Pi}^{(l-1)} \overset{d}{=} \Pi Z_X^{(l-1)}$ and $A \rightarrow \Pi A\Pi^T$, then the refined node representations $Z_{X,\Pi}^{(l)} \overset{d}{=} \Pi Z_X^{(l)}$. To prove this statement, we go through Algorithm 1 step by step.

- Step 2 implies $\tilde{Z}_{X,v,\Pi}^{(l-1)} \overset{d}{=} \tilde{Z}_{X,\pi(v)}^{(l-1)}$ because $\tau$ is element-wise operation.
- Steps 3: For both NeighborSample (categorical and Bernoulli) by Algorithm 2/3, the substeps 1-2 imply $\phi_{vt,\Pi}^{(l)} \overset{d}{=} \Pi \phi_{\pi(v)t}^{(l)}$. Here, we use $A \rightarrow \Pi A\Pi^T$ and thus $V_{vt} \rightarrow V_{\pi(v)t}$, and assume that $\phi_{vt,\Pi}^{(l)}, \phi_{\pi(v)t}^{(l)}$ are represented as vectors in $\mathbb{R}^{n\times 1}$ where their $u$th components, $\phi_{vt,\Pi,u}^{(l)}, \phi_{\pi(v)t,u}^{(l)}$, are 0's if $\pi^{-1}(u) \notin V_{vt}$. Substep 3, implies $Z_{A,v,\Pi}^{(l)} \overset{d}{=} \pi(Z_{A,\pi(v)}^{(l)})$ where $\pi(S) = \{\pi(v)|v \in S\}$ for some set $S \subseteq V$.
- Step 4 implies $\bar{Z}_{X,v,\Pi}^{(l)} \overset{d}{=} \bar{Z}_{X,\pi(v)}^{(l)}$.
- Steps 5-6 imply $\mu_{v,\Pi}^{(l)} \overset{d}{=} \mu_{\pi(v)}^{(l)}, \sigma_{v,\Pi}^{2(l)} \overset{d}{=} \sigma_{\pi(v)}^{2(l)}$.
- Step 7 implies $Z_{X,v,\Pi}^{(l)} \overset{d}{=} Z_{X,\pi(v)}^{(l)}$.

which indicates $Z_{X,\Pi}^{(l)} \overset{d}{=} \Pi Z_X^{(l)}$ and concludes the proof. $\qquad\square$

# F   Summary of the Datasets

Table 4 summarizes statistics of the datasets (Cora, Pubmed, Citeseer [43]) we use, as well as the standard train-validation-test split we use in the experiments.

Table 4: Summary of the datasets and splits in our experiments.

|  | Cora | Pubmed | Citeseer |
|---|---|---|---|
| **# Nodes** | 2708 | 19717 | 3327 |
| **# Edges** | 5429 | 44338 | 4732 |
| **# Features/Node** | 1433 | 500 | 3703 |
| **# Classes** | 7 | 3 | 6 |
| **# Training Nodes** | 140 | 60 | 120 |
| **# Validation Nodes** | 500 | 500 | 500 |
| **# Test Nodes** | 1000 | 1000 | 1000 |

# G   Implementation Details for the GIB-Cat, GIB-Bern and Other Compared Models

For all experiments and all models, the best models are selected according to the classification accuracy on the validation set. All models are trained with a total of 2000 epochs. For all experiments, we run it with 5 random seeds: 0, 1, 2, 3, 4 and report the average performance and standard deviation. The models are all trained on NVIDIA GeForce RTX 2080 GPUs, together with Intel(R) Xeon(R) Gold 6148 CPU @ 2.40GH CPUs. We use PyTorch [50] and PyTorch Geometric [51] for constructing the GNNs and evaluation. Project website and code can be found at http://snap.stanford.edu/gib/. In Section G.1, G.2 and G.3, we detail the hyperparameter setting for Section 5.1, and in Section G.4 and G.5, we provide additional details for the experiments.

## G.1   Implementation Details for the GIB-Cat and GIB-Bern

The architecture of GIB-Cat and GIB-Bern follows Alg. 1 (and Alg. 2 and 3 for the respective neighbor-sampling). We follow GAT [5]'s default architecture, in which we use 8 attention heads, nonlinear activation $\tau$ as LeakyReLU, and feature dropout rate of 0.6 between layers. We follow GAT's default learning rate, i.e. 0.01 for Cora and Citeseer, and $5 \times 10^{-3}$ for Pubmed. As stated in the main text, the training objective is Eq. (1), substituting in Eq. (5) and (6). To allow more flexibility (in similar spirit as $\beta$-VAE [41]), we allow the coefficient before $\widehat{\text{AIB}}$ and $\widehat{\text{XIB}}$ to be different, and denote them as $\beta_1$ and $\beta_2$. In summary, the objective is written as:

$$L = \sum_{v \in V} \text{Cross-Entropy}(Z_{X,v}^{(L)} W_{\text{out}}; Y_v) + \beta_1 \sum_{l \in S_A} \widehat{\text{AIB}}^{(l)} + \beta_2 \sum_{l \in S_X} \widehat{\text{XIB}}^{(l)} \tag{13}$$

In this work, we set the index set $S_A = [L] = \{1, 2, ...L\}$ and $S_X = \{L-1\}$, which satisfies Proposition 3.2. For $\widehat{\text{XIB}}$, we use mixture of Gaussians as the variational marginal distribution $\mathbb{Q}(Z_X)$. For the mixture of Gaussians, we use $m = 100$ components with learnable weights, where each component is a diagonal Gaussian with learnable mean and standard deviation. This flexible variational marginal allows it to flexibly approximate the true marginal distribution $\mathbb{P}(Z_X)$. For the reparameterization in $\widehat{\text{AIB}}$, we use Gumbel-softmax [24, 25] with temperature $\tau$. For GIB-Cat, the number of neighbors $k$ to be sampled is a hyperparameter. For GIB-Bern, we use Bernoulli($\alpha$) as the non-informative prior, where we fix $\alpha = 0.5$. To facilitate learning at the beginning, for the first 25% of the epochs we do not impose $\widehat{\text{AIB}}$ or $\widehat{\text{XIB}}$, and gradually anneal up both $\beta_1$ and $\beta_2$ during the 25% - 50% epochs of training, and keep them both at their final value afterwards. For the experiment in Section 5.1 and section 5.2, we perform hyperparameter search of $\beta_1 \in \{0.1, 0.01, 0.001\}$, $\beta_2 \in \{0.01, 0.1\}$, $\mathcal{T} \in \{1, 2\}$, $\tau \in \{0.05, 0.1, 1\}$, $k \in \{2, 3\}$ for each dataset, and report the one with higher validation F1-micro. A summary of the hyperparameter scope is in Table 5. In Table 6 and 7, we provide the hyperparameters that produce the results in Section 5.1, and in Table 8, we provide hyperparameters that produce the results in Section 5.2.

Table 5: Hyperparameter scope for Section 5.1 and 5.2 for GIB-Cat and GIB-Bern.

| Hyperparameters | Value/Search space | Type |
|---|---|---|
| $S_A$ | $[L]$ | Fixed* |
| $S_X$ | $\{L-1\}$ | Fixed |
| Number $m$ of mixture components for $\mathbb{Q}(Z_X)$ | 100 | Fixed |
| $\beta_1$ | $\{0.1, 0.01, 0.001\}$ | Choice† |
| $\beta_2$ | $\{0.1, 0.01\}$ | Choice |
| $\tau$ | $\{0.05, 0.1, 1\}$ | Choice |
| $k$ | $\{2, 3\}$ | Choice |
| $\mathcal{T}$ | $\{1, 2\}$ | Choice |

*Fixed: a constant value

†Choice: choose from a set of discrete values

Table 6: Hyperparameter for adversarial attack experiment for GIB-Cat and GIB-Bern.

| Dataset | Model | $\beta_1$ | $\beta_2$ | $\tau$ | $k$ | $\mathcal{T}$ |
|---|---|---|---|---|---|---|
| Cora | GIB-Cat | 0.001 | 0.01 | 1 | 3 | 2 |
| | GIB-Bern | 0.001 | 0.01 | 0.1 | - | 2 |
| Pubmed | GIB-Cat | 0.001 | 0.01 | 1 | 3 | 2 |
| | GIB-Bern | 0.001 | 0.01 | 0.1 | - | 2 |
| Citeseer | GIB-Cat | 0.001 | 0.01 | 0.1 | 2 | 2 |
| | GIB-Bern | 0.001 | 0.01 | 0.05 | - | 2 |

Table 7: Hyperparameter for adversarial attack experiment for the ablations of GIB-Cat and GIB-Bern.

| Model | $\beta_1$ | $\beta_2$ | $\tau$ | $k$ | $\mathcal{T}$ |
|---|---|---|---|---|---|
| AIB-Cat | - | 0.01 | 1 | 3 | 2 |
| AIB-Bern | - | 0.01 | 0.1 | - | 2 |
| XIB | 0.001 | - | - | - | 2 |

Table 8: Hyperparameter for feature attack experiment (Section 5.2) for GIB-Cat and GIB-Bern.

| Dataset | Model | $\beta_1$ | $\beta_2$ | $\tau$ | $k$ | $\mathcal{T}$ |
|---|---|---|---|---|---|---|
| Cora | GIB-Cat | 0.01 | 0.01 | 0.1 | 2 | 2 |
| | AIB-Cat | - | 0.01 | 0.1 | 2 | 2 |
| | GIB-Bern | 0.001 | 0.01 | 0.05 | - | 2 |
| | AIB-Bern | - | 0.01 | 0.05 | - | 2 |
| Pubmed | GIB-Cat | 0.001 | 0.01 | 1 | 3 | 2 |
| | AIB-Cat | - | 0.01 | 1 | 3 | 2 |
| | GIB-Bern | 0.01 | 0.01 | 0.05 | - | 1 |
| | AIB-Bern | - | 0.01 | 0.05 | - | 1 |
| Citeseer | GIB-Cat | 0.001 | 0.01 | 0.1 | 2 | 2 |
| | AIB-Cat | - | 0.01 | 0.1 | 2 | 2 |
| | GIB-Bern | 0.1 | 0.01 | 0.05 | - | 2 |
| | AIB-Bern | - | 0.01 | 0.05 | - | 2 |

## G.2 Implementation Details for GCN and GAT

We follow the default setting of GCN [3] and GAT [5], as implemented in https://github.com/rusty1s/pytorch_geometric/blob/master/examples/gcn.py and https://github.com/rusty1s/pytorch_geometric/blob/master/examples/gat.py, respectively. Importantly, we keep the dropout on the attention weights as the original GAT. Whenever possible, we keep the same architecture choice between GAT and GIB-Cat (and GIB-Bern) as detailed in Section G.1, for a fair comparison.

### G.3 Implementation Details for RGCN and GCNJaccard

We used the implementation in this repository: `https://github.com/DSE-MSU/DeepRobust`. We perform hyperparameter tuning for both baselines for the adversarial attack experiment in Section 5.1. We first tune the latent dimension, learning rate, weight decay for both models. Specifically, we search within $\{16, 32, 64, 128\}$ for latent dimension, $\{10^{-3}, 10^{-2}, 10^{-1}\}$ for learning rate, and $\{10^{-4}, 5 \times 10^{-4}, 10^{-3}\}$ for weight decay. For GCNJaccard, we additionally fine-tune the threshold hyperparameter which is used to decide whether two neighbor nodes are still connected. We search threshold within $\{0.01, 0.03, 0.05\}$. For RGCN, we additionally fine-tune the $\beta_1$ within $\{10^{-4}, 5 \times 10^{-4}, 10^{-3}\}$ and $\gamma$ within $\{0.1, 0.3, 0.5, 0.9\}$. Please find the best set of hyperparameters for both models in Table 9, 10 and 11.

Table 9: Hyperparameter of baselines used on Citeseer dataset.

|  | RGCN | GCNJaccard |
|---|---|---|
| latent dim | 64 | 16 |
| learning rate | $10^{-2}$ | $10^{-2}$ |
| weight dacay | $5 \times 10^{-4}$ | $5 \times 10^{-4}$ |
| threshold | - | $5 \times 10^{-2}$ |
| $\beta_1$ | $5 \times 10^{-4}$ | - |
| $\gamma$ | 0.3 | - |

Table 10: Hyperparameter of baselines used on Cora dataset.

|  | RGCN | GCNJaccard |
|---|---|---|
| latent dim | 64 | 16 |
| learning rate | $10^{-2}$ | $10^{-2}$ |
| weight dacay | $5 \times 10^{-4}$ | $5 \times 10^{-4}$ |
| threshold | - | $5 \times 10^{-2}$ |
| $\beta_1$ | $5 \times 10^{-4}$ | - |
| $\gamma$ | 0.3 | - |

Table 11: Hyperparameter of baselines used on Pubmed dataset.

|  | RGCN | GCNJaccard |
|---|---|---|
| latent dim | 16 | 16 |
| learning rate | $10^{-2}$ | $10^{-2}$ |
| weight dacay | $5 \times 10^{-4}$ | $5 \times 10^{-4}$ |
| threshold | - | $5 \times 10^{-2}$ |
| $\beta_1$ | $5 \times 10^{-4}$ | - |
| $\gamma$ | 0.1 | - |

### G.4 Additional Details for Adversarial Attack Experiment

We use the implementation of Nettack [15] in the repository `https://github.com/DSE-MSU/DeepRobust` with default settings. As stated in the main text, for each dataset we select 40 nodes in the test set to attack with 10 having the highest margin of classification, 10 having the lowest margin of classification (but still correctly classified), and 20 random nodes. For each target node, we independently train a different model and evaluate its performance on the target node in both evasive and poisoning setting. Different from [15] that only keeps the largest connected component of the graph and uses random split, to keep consistent settings across experiments, we still use the full graph and standard split, which makes the defense even harder than that in [15]. For each dataset and each number of perturbations (1, 2, 3, 4), we repeat the above experiment 5 times with random seeds 0, 1, 2, 3, 4, and report the average accuracy on the targeted nodes (therefore, each cell in Table 1 is the mean and std. of the performance of 200 model instances (5 seeds × 40 targeted nodes, each training one model instance). Across the 5 runs of the experiment, the 20 nodes with highest and lowest margin of classification are kept the same, and the 20 random nodes are sampled randomly

Table 12: Average classification accuracy (%) for the targeted nodes under direct attack for Cora. Each number is the average accuracy for the 40 targeted nodes for 5 random initialization of the experiments. Bold font denotes top two models.

| | Clean (%) | Evasive (%) | | | | Poisoning (%) | | | |
|---|---|---|---|---|---|---|---|---|---|
| | | **1** | **2** | **3** | **4** | **1** | **2** | **3** | **4** |
| DGI | $83.2_{\pm 4.82}$ | $54.5_{\pm 4.81}$ | $41.5_{\pm 2.24}$ | $35.5_{\pm 5.42}$ | $31.0_{\pm 3.79}$ | $53.5_{\pm 7.42}$ | $38.5_{\pm 4.18}$ | $33.0_{\pm 5.42}$ | $29.0_{\pm 3.79}$ |
| **GIB-Cat** | $77.6_{\pm 2.84}$ | $\mathbf{63.0}_{\pm 4.81}$ | $\mathbf{52.5}_{\pm 3.54}$ | $\mathbf{44.5}_{\pm 5.70}$ | $\mathbf{36.5}_{\pm 6.75}$ | $\mathbf{60.0}_{\pm 6.37}$ | $\mathbf{50.0}_{\pm 2.50}$ | $\mathbf{39.5}_{\pm 5.42}$ | $\mathbf{30.0}_{\pm 3.95}$ |
| **GIB-Bern** | $78.4_{\pm 4.07}$ | $\mathbf{64.0}_{\pm 5.18}$ | $\mathbf{51.5}_{\pm 4.54}$ | $\mathbf{43.0}_{\pm 3.26}$ | $\mathbf{37.5}_{\pm 3.95}$ | $\mathbf{61.5}_{\pm 4.18}$ | $\mathbf{46.0}_{\pm 4.18}$ | $\mathbf{36.5}_{\pm 4.18}$ | $\mathbf{31.5}_{\pm 2.85}$ |

and then fixed. We also make sure that for the same seed, different models are evaluated against the same 40 target nodes, to eliminate fluctuation between models due to random sampling.

### G.5 Additional Details for Feature Attack Experiment

As before, for each model to compare, we train 5 instances with seeds 0, 1, 2, 3, 4. After training, for each seed and each specified feature noise ratio $\lambda$, we perform 5 random node feature attacks, by adding independent Gaussian noise $\lambda \cdot r \cdot \epsilon$ to each dimension of the node feature, where $r$ is the mean of the maximum value of each node's feature, and $\epsilon \sim N(0, 1)$. Therefore, each number in Table 3 is the mean and std. of 25 instances (5 seeds $\times$ 5 attacks per seed).

## H   Training time for GIB-Cat and GIB-Bern

The training time of GIB-Cat and GIB-Bern is on the same order as GAT with the same underlying architecture. For example, with 2 layers, GIB-Cat takes 98s (GIB-Bern takes 84s) to train 2000 epochs on a NVIDIA GeForce RTX 2080 GPU, while GAT takes 51s to train on the same device. The similar order of training time is due to that they have similar number of parameters and complexity. Compared to GAT, GIB-Cat and GIB-Bern introduce minimal more parameters. In this work, on the structural side, we use the attention weights of GAT as parameters to encode structural representation, which keeps the same number of parameters as GAT. On the feature side, we set $S_X = \{L - 1\}$, which only requires to predict the diagonal variance of the Gaussian in addition to the mean, which introduce small number of parameters. Therefore, in total, GIB-Cat and GIB-Bern have similar complexity. The added training time is due to the sampling of edges and node features during training. We expect that when GIB is applied to other GNNs, the augmented model has similar complexity and training time.

## I   Additional experiments for Deep Graph Infomax (DGI)

Here we perform additional experiment for adversarial attacks on Cora using Nettack. The result is in Table 12. We see that both GIB-Cat and GIB-Bern outperform DGI by a large margin.

## J   More Detailed Analysis of Adversarial Attack in Section 5.1

Table 13 summarizes the statistics of the target nodes and the adversarial perturbations by Nettack, for Cora, Pubmed and Citeseer.

Table 13: Statistics of the target nodes and adversarial perturbations by Nettack in Section 5.1.

| | **Cora** | **Pubmed** | **Citeseer** |
|---|---|---|---|
| Fraction of degree 1 in target nodes | 0.215 | 0.425 | **0.500** |
| Fraction of degree $\leq 2$ in target nodes | 0.345 | 0.565 | **0.710** |
| Fraction of degree $\leq 3$ in target nodes | 0.455 | 0.630 | **0.755** |
| Fraction of degree $\leq 4$ in target nodes | 0.540 | 0.640 | **0.810** |
| Fraction of structural attacks | 1.000 | 1.000 | 0.991 |
| Fraction of added-edge attack in structural attacks | 0.890 | 0.834 | 0.909 |
| Fraction of different classes in added-edge attacks | 1.000 | 0.993 | 0.985 |

We have the following observations:

- Compared to Cora and Pubmed, Citeseer has much more nodes with degrees less than 1, 2, 3, 4. This explains why in general the 5 models has worse performance in Citeseer than in Cora and Pubmed.

- Almost all attacks ($\geq 99.1\%$) are structural attacks.

- Within structural attacks, most of them ($\geq 83.4\%$) are via adding edges, with Citeseer having the largest fraction.

- For the added edges, almost all of them ($\geq 98.5\%$) have different classes for the end nodes.

From the above summary, we see that the target nodes in Citeseer dataset in general have fewest degrees, which are most prone to added-edge structural attacks by connecting nodes with different classes. This exactly satisfies the assumption of GCNJaccard [34]. GCNJaccard proceeds by deleting edges with low feature similarity, so those added edges are not likely to enter into the model training during poisoning attacks. This is probably the reason why in Nettack poisoning mode in Citeseer, GCNJaccard has the best performance.