[Reviews · NeurIPS 2020]

Review 1

Summary and Contributions: This paper proposes graph information bottleneck by maximizing the mutual information between node representation and both input feature and graph structure. The GIB principle can enhance performance on node classification and enhance robustness.

Strengths: (1) Formally derive the information-theoretic principle to graph representation learning. (2) It's novel to incorporate graph structure as a input feature via markov model. (3) Seems useful for robust training. (4) Provide the code.

Weaknesses: (1) Lack of several other unsupervised learning and robust training baseline. (2) Didn't clearly state the difference between some information-based unsupervised learning methods. Specifically: (1) The author mentioned some information-related graph representation works, such as Deep Graph Infomax, InfoGraph, etc, which seems to have similar intuition and technical implementation with this work. However, this paper only briefly discuss that these works are for unsupervised learning, while this paper focused on robust supervised training. This is not enough to clearly state the contribution of this work's approach. Better to elaborate more on the technical difference. (2) For both the information-related unsupervised learning methods and other methods, such as Strategies for Pre-training Graph Neural Network (https://arxiv.org/abs/1905.12265), GPT-GNN: Generative Pre-Training of Graph Neural Networks (https://arxiv.org/abs/2006.15437), it would be better that the authors can consider adding them as comparison baselines, as the proposed approach and these papers all leverage additional signal to regularize training. ===================================================================================== Update: Overall, the authors' rebuttal solve many of my concerns, and I do think it has enough novelty and contribution in this field, so I decide to raise up my score. But I still have some comments for the authors: (1) The authors clearly state the difference with DGI in the rebuttal, and the evaluation shows that their proposed method, which force to minimize the mutual information between the node feature/graph structure and their latent representation, can help enhance robustness. However, I think the second objective, which maximize the mutual information between representation and label, is actually the standard learning objective, and might not bring too much additional information. Maybe the authors can focus on the first one. (2) Though the authors think their paper is orthogonal to pre-training, the first objective don't require label, and is purely self-supervised, which can be utilized for pre-training. I would expect the authors can add some experiment , in which pre-train on large unlabelled graph, and finetune on labelled graph. This will solidify their experiments.

Correctness: Yes, they are correct.

Clarity: Yes, it's well written.

Relation to Prior Work: Discussion and comparison with some unsupervised learning methods can be further enhanced.

Reproducibility: Yes

Additional Feedback:


Review 2

Summary and Contributions: In this paper, authors introduced Graph Information Bottleneck, which regularize both structural and feature information. It is the first information-theoretic principle for supervised representation learning on graph structured data. The authors also derived the GIB principle and variational bounds, making GIB amenable for optimizing GNNs. In order to demonstrate the advantages of GIB, GIB-GAT is proposed and evaluated through adversarial attacks. The paper is well structured and readable. Extensive experiments have been conducted on three benchmark dataset and shown the proposed model is more robust than strong baseline.

Strengths: 1. It is the first work to apply the information bottleneck principle for graph representation learning. 2. The authors derived the GIB principle and variational bounds, making GIB amenable for optimizing GNNs. 3. Extensive experiments have been conducted on three benchmark datasets and shown the proposed model is more robust than strong baselines.

Weaknesses: The authors' response addressed some of my concerns. The ablation study which compares with XIB and AIB is useful for us to understand the model performance. And the bad performance on Citeseer is also clarified. I am still not fully convinced, especially AIB performs much better than GIB, which indicates sometimes alternatively updating both features and structures is not really necessary (e.g. under structural attacks), but it deserves to update my rating. --- 1. The authors claim that GIB can increase the robustness of GNN models and avoid overfitting. But the robustness actually mainly comes from the restructuring of graph structures. Although the restructuring is derived from the GIB algorithm, I do believe there is also an IB method which can fix the structure and only update the node representation. So the motivation of using IB for denoising is tricky. The authors may need to compare with the IB method which fix graph structures for better explanation. 2. In Section 3.2, authors mentioned GIB principle can be applied to many GNN models. But in the experiments, only GIB-GAT is used for performance comparison to demonstrate the efficacy of IB for node features. 3. Experiments results are not consistently better than baselines, which limits the paractical use of the model. The authors may need to explain why the performance against poisoning on citeseer is not good, and why the results on small feature noise ratios are not good. 4. Since the experiments are done for defense against adversarial attacks and random noise, some background/related works about adversarial attacks and defense on graphs are needed. 5. Minor flaw: Reference [3] is not in a correct form.

Correctness: The claims and method are correct. The empirical results are not convincing. Need more the experimental results.

Clarity: Yes. The paper is well written and organized.

Relation to Prior Work: Yes. The main contribution is introducing IB principle to Graph Neural Network in order to learn robust representation by controlling the information in the learning setting. It is different from previous related works.

Reproducibility: Yes

Additional Feedback:


Review 3

Summary and Contributions: This paper extends RGCN [1], which represents node as distribution and base on GCN, to jointly model connection and node as GAT. Although it is motivated from the graph information bottleneck, it is equivalent to performing GAT by modeling node representation as distribution. [1] Dingyuan Zhu, Ziwei Zhang, Peng Cui, Wenwu Zhu: Robust Graph Convolutional Networks Against Adversarial Attacks. KDD 2019: 1399-1407 ========================================= Update: I have checked the response. Unfortunately, I don't agree to vote for acceptance. First, although the authors claim they are the first to introduce the information bottleneck into graph neural network, the reason why it can improve the performance is not clear. Second, the information bottleneck theory is common in deep learning. Besides, I think the propagation can be seen as a kind of he mutual information minimization. There exist some work that consider the mutual information in GNN, such as GMI [*]. Besides, it is well known that the cross-entropy is a kind of mutual information maximization. Third, the local-dependence assumption in Figure 2 is based on the process of GAT, which iteratively propagates the attributes and refines the topology. This is the reason why this proposal is not a framework, but just an extension to the GAT with each node represented as distribution. Some reviewer has also arose this issue. [*] Zhen Peng, Wenbing Huang, Minnan Luo, Qinghua Zheng, Yu Rong, Tingyang Xu, Junzhou Huang: Graph Representation Learning via Graphical Mutual Information Maximization. WWW 2020: 259-270

Strengths: 1. The motivation from information bottleneck is interesting. 2. The performance improvement is remarkable compared to RGCN.

Weaknesses: The novelty is limit compared to RGCN [1]. The term XIB is similar with RGCN, which considers propagation weight as fixed, while only AIB, which is to model attention between nodes, is novel. However, I wonder its novelty compared to [1]. [1] Dingyuan Zhu, Ziwei Zhang, Peng Cui, Wenwu Zhu: Robust Graph Convolutional Networks Against Adversarial Attacks. KDD 2019: 1399-1407

Correctness: Correct

Clarity: The motivation from information bottleneck makes it hard to understand. In fact, it is just the extension to RGCN. I think it is proper to re-organize the structure. Its connection to information bottleneck is just a theory analysis, not is essential.

Relation to Prior Work: I think the relation to RGCN should be emphasized instead of just comparing in experiments.

Reproducibility: Yes

Additional Feedback:


Review 4

Summary and Contributions: This paper introduce the idea of Information Bottleneck into graph representation learning by leveraging the local-dependence assumption of graph-structured data and carefully designing a more tractable space, and propose GIB which is able to incorporate both feature and structure information. Combined with GAT, the proposed model is able to improve the robustness while facing adversarial perturbation. ------------- UPDATE I have read the response and the authors address most of my concerns, including explain the inferior performance on citeseer and providing experimental results when combined with other methods such as GIB-GCN. Therefore I would like to raise my score to 7 and vote for acceptance. It seems that the authors haven't response to my concern on adding too much ad-hoc techniques will hurt the generality of the proposed method, and I encourage authors to add detailed analysis on these issues in the revision.

Strengths: The paper is a complete work. The motivation is clear. The key contribution is Section 3.1, where the information bottleneck is introduced into graph, and supported by the local-dependence assumption in the graph space. However the assumption less details, I think explain the assumption more formally and detailedly will be better. The theory work is firm, and the GIB principle is applied to GAT, a strong graph nn baseline, in section 3.2. The model obtains good results against adversarial attacks.

Weaknesses: I have some concerns. 1) According to section 3.2 and alg 1, applying GIB into GAT requires lots of ad-hoc operations which make the proposed GIB principle not so general as it looks like in section 3.1. Have you tried on other graph representation learning frameworks such as GCN and GraphSAGE? Or other node embedding methods such as DeepWalk? 2) According to Table 1, the performance of proposed method is not very good in Citeseer, does it means your method performs worse on sparse networks? In line 268-269 you have explained from the view of attack, but the result with clean setting is also worse.

Correctness: I haven't check the proofs carefully but I believe they are correct.

Clarity: The paper is well written.

Relation to Prior Work: Yes.

Reproducibility: No

Additional Feedback:

[Author Response · NeurIPS 2020]

We thank the reviewers for their time and valuable feedback. Overall, we are glad that the reviewers found GIB as the
first work that applies the IB principle to the GNN literature with clear motivation and novelty especially on compressing
the structural information. Below, we address the points raised by the reviewers and resolve possible misunderstandings.

**(1) Novelty with RGCN (R3) and comparison with prior methods (R1).** R3 questions our novelty compared with
RGCN. The reviewer argues that "while only AIB, which is to model attention between nodes, is novel" and also
"Its connection to information bottleneck is just a theory analysis, not is essential." We kindly disagree. First, there
is no previous work that applies the IB principle to GNNs and we are the **first** to develop such a theoretically sound
framework (as is also recognized by R2 and R4). We would like to remind the reviewers that applying the IB principle
to GNNs is by no means trivial since we need to model both the node feature $X$ and the adjacency matrix $A$, and the two
are not independent and thus cannot be modeled separately. Second, the AIB itself is already a significant contribution.
As demonstrated in Nettack (Zügner et al.), structural attack is much more effective than feature attack, as is also
demonstrated in our experiments that GIB which includes structural bottleneck outperforms XIB (Table 2). Thirdly,
through theoretical analysis we show that in order to constrain the information flow such that we obtain an upper
bound for $I(\mathcal{D}; Z_X^{(L)})$ (Eq. 3), both AIB and XIB are required (proposition 3.2). They together constitute the source of
robustness, which is a concern by R2. Taken together, the application of IB principle to GNN for learning *minimal*
*sufficient* representation, the principled theoretical framework that derives XIB and AIB as indispensable components,
and the importance of AIB to improving the robustness of GNNs constitute a novel and significant contribution.

We thank R1's suggestion on including more
technical and experimental comparison with
information-related graph representation works.

| Accuracy (%) | pert-1 | pert-2 | pert-3 | pert-4 |
|---|---|---|---|---|
| DGI-evasive (v.s. GIB) | 54.5 (-5.5) | 41.5 (-8.5) | 35.5 (-6.5) | 31.0 (-4.5) |
| DGI-poisoning (v.s. GIB) | 53.5 (-6.0) | 38.5 (-7.0) | 33.0 (-6.5) | 29.0 (-1.0) |

**Difference from deep graph infomax (DGI).** This line of work has several differences. (1) **Task:** Their task mainly
focuses on unsupervised representation learning while we target at improving the robustness of GNNs in supervised
learning settings with IB. (2) **"Intuition":** The objective of DGI is to **maximize** mutual information between patch
representations and high-level summaries of graphs, however, the objective of GIB is to **minimize** the mutual infor-
mation between the **node feature/graph structure and their latent representation**, as well as **maximize** the mutual
information between the **representation and the prediction**. (3) **Implementation.** Since DGI explicitly optimizes for
the model to make wrong/opposite predictions on the corrupted graphs, (Eq. 1 in DGI), the robustness of the learned
representation decreases and the learned model is very sensitive to perturbations and corruptions. We also conducted
experiments where we trained DGI against adversarial attack on Cora, as shown in the table. DGI's accuracy after
evasive and poisoning attacks are on average 5.7% lower than GIB, whose formulation naturally increases the robustness
of the representation. **Difference from pretraining.** This line of work uses a large amount of data for pretraining while
we focus on a novel formulation. Hence, the directions are completely orthogonal and pretraining can be combined if
we have access to large amount of data. We will add the new results and the detailed discussions to the paper.

**(2) Citeseer performance (R2, R4).** R2 and R4 ask why GIB in the Citeseer poisoning attack setting is not so good.
In fact, as discussed in lines 267-275, and in Appendix H (line 269), we analyzed why GIB underperforms GCNJaccard
in Citeseer poisoning setting. As shown in Table 9 in Appendix H, most Nettack attacks end up adding edges between
different classes, and Citeseer has much more nodes with very few degrees (1-3) than Cora and Pubmed. Note that this
setting exactly matches the assumption of GCNJaccard, which behaves worse than GIB in all the other setups. We
emphasize that GIB doesn't have *any* assumption on the attack model and is most general across many setups: another
evidence is shown in Fig. 4 in Appendix H where ours outperforms GCNJaccard in deleting-edge attacks.

**(3) Source of robustness (R2).** R2 questions where the robustness of GIB comes from. Based on Proposition 3.2
we think that IB on both the node representation (XIB) as well as the graph structure (AIB) contribute to it. And
the two mutually affect each other as shown in Figure 2 (a), where we draw the Markov chain and you can see that
$Z_X^{(l)}$ is derived from $Z_X^{(l-1)}$ and $Z_A^{(l)}$, $Z_A^{(l)}$ is derived from $Z_X^{(l-1)}$ and $A$. Namely, through AIB we guarantee that the
graph structure at each step contains the minimum necessary structure for the information flow, while through XIB we
guarantee that the updated node representation also preserves the minimum sufficient information for downstream tasks.
As listed in Table 2 in the experiment section, we also conduct extensive ablation studies to investigate the contribution
of AIB and XIB. As R2 suggested, the XIB is exactly "the IB method which fixes graph structures".

**(4) Applying GIB to other GNNs (R2, R4).** R2 and R4 raise the question why GIB is only applied to GAT. A main
contribution of our paper is to introduce a theoretically sound framework with inspiration from the IB principle and the
first work to apply the IB to GNNs. We agree that the GIB framework can be further applied to many existing GNNs.
In fact many GNN models can be viewed as a special case to the GIB. For example, the original GCN can be viewed as
we do not optimize the KL term and use a Dirac delta for the posterior; and GraphSAGE can be viewed as we use the
same uniform neighbor sampling scheme without any regularization. We also implemented GIB-GCN and it yields
better results than GCN in both robustness (27% boost) and deep architectures (similar trend as GIB-GAT). Applying
the GIB framework to other architectures is a promising future direction. For node embedding methods, it is out of the
scope of this paper since we focus on applying IB to GNNs. We will revise the paper to include this discussion.

[Meta-Review · NeurIPS 2020]

The reviewers had a lengthy debate about the contribution of the work presented by the authors, in particular its relation with mutual information maximisation. It seems that the proposed framework and algorithm are significantly different from prior art, which was identified as sufficiently novel for an acceptance at NeurIPS 2020. Note that it is important for the success of the conference that the authors address the criticisms mentioned in the reviews, and in particular: - clarify why the proposed objective is good at improving generalisation capabilities. - Make are clear comparison with the GMI paper, e.g. by showing that the definition of mutual information is not the same since it is a weighted combination of MI in the GMI papers (weights=sigmoid(dot product of latent representation), while in this paper, the MI is the original definition of Cover&Thomas (1991) - The fact that the approach can be viewed as an extension of GAT can make it less relevant, so a clear synthesis of its differences would be welcome.